# Electronic and Magnetic Properties of Building Blocks of Mn and Fe Atomic Chains on Nb(110)

**DOI:** 10.3390/nano11081933

**Published:** 2021-07-27

**Authors:** András Lászlóffy, Krisztián Palotás, Levente Rózsa, László Szunyogh

**Affiliations:** 1Wigner Research Centre for Physics, Institute for Solid State Physics and Optics, H-1525 Budapest, Hungary; laszloffy.andras@wigner.hu; 2Department of Theoretical Physics, Budapest University of Technology and Economics, H-1111 Budapest, Hungary; szunyogh.laszlo@ttk.bme.hu; 3MTA-SZTE Reaction Kinetics and Surface Chemistry Research Group, University of Szeged, H-6720 Szeged, Hungary; 4Department of Physics, University of Konstanz, D-78457 Konstanz, Germany; levente.rozsa@uni-konstanz.de; 5MTA-BME Condensed Matter Research Group, Budapest University of Technology and Economics, H-1111 Budapest, Hungary

**Keywords:** magnetism, Ab initio, spin model, isotropic interaction, embedded cluster, adatom, dimer, monatomic chain, Mn, Fe, Nb

## Abstract

We present results for the electronic and magnetic structure of Mn and Fe clusters on Nb(110) surface, focusing on building blocks of atomic chains as possible realizations of topological superconductivity. The magnetic ground states of the atomic dimers and most of the monatomic chains are determined by the nearest-neighbor isotropic interaction. To gain physical insight, the dependence on the crystallographic direction as well as on the atomic coordination number is analyzed via an orbital decomposition of this isotropic interaction based on the spin-cluster expansion and the difference in the local density of states between ferromagnetic and antiferromagnetic configurations. A spin-spiral ground state is obtained for Fe chains along the [$1\bar{1}0$] direction as a consequence of the frustration of the isotropic interactions. Here, a flat spin-spiral dispersion relation is identified, which can stabilize spin spirals with various wave vectors together with the magnetic anisotropy. This may lead to the observation of spin spirals of different wave vectors and chiralities in longer chains instead of a unique ground state.

## 1. Introduction

Over the past decades, the exploration of exotic magnetic patterns in nanostructures has become an active research field. Beside significant advances in molecular metallic chains, so-called single chain magnets [1,2,3], this area has recently extended towards the investigation of magnetic-superconducting heterostructures, which may find applications in quantum computing. The presence of magnetic impurities on a superconducting surface leads to the emergence of so-called Yu–Shiba–Rusinov (YSR) states [4,5,6,7,8,9], due to the coupling of the localized magnetic moment to the Cooper pairs. In atomic chains, hybridized YSR states develop into bands [10,11], that can give rise to topological superconductivity and consequently to the appearance of Majorana bound states [12,13,14,15,16]. In a recent study by Beck et al. [7], an Mn adatom and Mn dimers on Nb(110) were investigated in the superconducting state. The experimental results supported by tight-binding model calculations based on density functional theory (DFT) data demonstrated that the YSR states hybridize not only for ferromagnetic but also for antiferromagnetic dimers, as a consequence of the spin–orbit coupling (SOC) present in the system.

In atomic chains, the magnetic structure has a profound effect on the emergence of topological superconductivity, with a spin-spiral ground state having been identified as a key element in finding the Majorana bound states at the ends of the chain [17,18,19,20]. The spin spiral state may be formed by the Dzyaloshinskii–Moriya interaction [21,22] or by the frustration of the isotropic interactions, as it has been found in magnetic thin films [23,24] and atomic chains [25] based on *ab initio* calculations.

Taking into account the frustration of isotropic interactions necessitates going beyond a nearest-neighbor approximation, and they are often essential for determining the ground state of spin systems, a very recent example is shown for Mn atomic chains on W(110) [26]. In [24], the second-nearest-neighbor (2NN) interactions play an important role in the wave number and also in the tilting of the spin-spiral ground state of the Re/Co/Pt(111) system. Even longer-ranged interactions—up to fifth neighbors—are necessary to be taken into account for Fe chains on Re(0001) for finding the spin-spiral ground state [25]. If the Fourier transform of the isotropic interactions has pockets with flat dispersion relation in the Brillouin zone, magnetic domains with different wave vectors can form. This was recently demonstrated by Kamber and coworkers [27] on the (0001) surface of crystalline Nd. Together with the aging effect—where the magnetic state depends on its history, this points toward the formation of a spin glass state, usually observed in disordered materials rather than crystalline systems with long-range periodicity.

As discussed above, determining the magnetic configuration is of crucial importance to explain the subgap fermionic states in magnetic-superconducting hybrid systems. In the present paper, we investigate the magnetic properties of Mn and Fe adatoms, dimers, and monatomic chains on Nb(110) along different crystallographic directions. Mn and Fe adatoms, Mn dimers, and Mn chains have already been studied on Nb(110) [7,8,14,28], the magnetic properties of which are now systematically investigated from the theoretical side. Determining the magnetic properties of atomic chains from first principles is a long-standing challenge [25,29,30,31,32]. The calculations here are carried out in the non-superconducting state, assuming that superconductivity does not affect the magnetic pattern; this is supported by the fact that the magnetic interactions are usually at least two orders of magnitude stronger than the superconducting gap. We also explore the connection between the local density of states and the decomposition of the isotropic Heisenberg exchange interaction into atomic orbitals, for different relative alignments and coordination numbers of the atoms. While most chains are found to be collinear ferromagnetic or antiferromagnetic, Fe chains on Nb(110) along the [$1\bar{1}0$] direction exhibit a spin-spiral ground state with a flat dispersion relation due to the frustration of the isotropic couplings, similar to the foundings for Nd in [27].

The paper is organized as follows. In Section 2.1 and Section 2.2, the details of the *ab initio* calculations are discussed, performed using the Vienna Ab initio Simulation Package (vasp) [33,34,35] to determine the equilibrium atomic geometry, and the embedding technique within the Korringa–Kohn–Rostoker (KKR) method [36], respectively. In Section 2.3, the classical spin model is introduced. The results for magnetic adatoms, dimers, and chains are discussed in Section 3.1, Section 3.2 and Section 3.3, respectively. The results are summarized in Section 4. Some analytical methods for determining the ground state angle of magnetic dimers are given in Appendix A.

## 2. Computational Details

The electronic and magnetic structure of the considered and similar metallic systems can usually be well described by using the local density approximation (LDA) [9,10,26,27] or the generalized gradient approximation (GGA) [8] within DFT.

### 2.1. VASP Calculations

The lowest-energy atomic geometries of the Mn and Fe adatoms on the Nb(110) surface were determined separately by using the Vienna Ab initio Simulation Package (VASP) [33,34,35]. The considered supercells included a single Mn or Fe adatom deposited in a hollow position on a four atomic layer-thick Nb slab, with 7×7 atoms in each layer in the bcc(110) geometry ensuring a minimal distance of 2 nm between the magnetic adatoms in repeating supercells to avoid artificial interaction among them. The bulk lattice constant of $a_{\mathrm{Nb}}$ = 330.04 pm was considered. The GGA for the exchange-correlation potential was used as parametrized by Perdew, Burke and Ernzerhof [37] similarly as in [8], and the Brillouin zone was sampled by the Γ point only, due to the large size of the supercell. A vacuum region of more than 1 nm was considered in the surface normal [110] direction to avoid unphysical interactions between artificially repeated slabs inherent to the supercell method. During the calculations, the Nb atoms in the top layer and the magnetic adatom were allowed to relax in the vertical direction only, keeping their lateral positions. The equilibrium atomic positions were found by minimizing the total energy. As a result, for the Mn/Nb system, the average vertical distance between the atoms in the two highest Nb layers decreased to 227.49 pm from the bulk value of $a_{\mathrm{Nb}}\sqrt{2}/2$ = 233.37 pm, and the Mn adatom relaxed even further, to a vertical distance of 198.68 pm measured from its nearest-neighbor (NN) Nb atoms. The spin magnetic moment of the Mn adatom was found to be 3.60 µ${}_{B}$. Note that the geometry of the Mn adatom is identical to the one used in [7]. For the Fe/Nb system, the average vertical distance between the atoms in the two highest Nb layers decreased to 225.42 pm, and the Fe adatom relaxed to a vertical distance of 174.95 pm measured from its NN Nb atoms (170 pm in [8]). The spin magnetic moment of the Fe adatom was found to be 2.14 µ${}_{B}$ (2.2 µ${}_{B}$ in [8]). These atomic geometries were used in the subsequent KKR calculations, where the vertical distance between the NN Nb and the respective magnetic adatom was also considered between the first two vacuum layers.

### 2.2. KKR Calculations

We used the Green’s function embedding technique based on the KKR multiple scattering theory [36] to determine the electronic and magnetic properties of the Mn and Fe atomic clusters. The Nb(110) surface has been modeled as an interface region between semi-infinite bulk Nb and vacuum, consisting of eight atomic layers of Nb and four atomic layers of empty spheres (vacuum). The energy integrals were performed using 16 points along a semicircle contour in the upper complex semi-plane. A sampling of up to 1012 $\vec{k}$ points in the Brillouin zone was used to do the self-consistent field (SCF) calculations and for the adatoms, and 7812 $\vec{k}$ points were used to calculate the Green’s function of the host for magnetic dimers and chains. The Ceperley–Alder-type of exchange-correlation functionals [38] as parametrized by Vosko, Wilk, and Nusair [39], and an angular momentum cut-off of $l_{\max}=3$ were considered in the KKR calculations. A single Mn or Fe adatom, dimers and chains consisting of 5, 10, and 15 Mn or Fe atoms were calculated by embedding them in the first vacuum layer with the layer relaxations described in Section 2.1.

We used the spin cluster expansion (SCE) [40,41,42] to investigate the magnetic interactions in the systems, similarly to the work in [43]. The spin model is discussed in Section 2.3. Alternatively, some of the model parameters may be estimated by calculating energy differences between specific magnetic configurations. In the spirit of the magnetic force theorem (MFT) [44], these were determined with fixed electronic potentials based on the band energy, which is obtained in two ways: either via direct integration of the local density of states (LDOS), or by using Lloyd’s formula [45,46].

First, we performed self-consistent calculations for the Mn and for the Fe adatom on the top of the Nb(110) substrate with the embedded cluster KKR technique. We considered clusters of different atoms and concluded that the spin magnetic moment of Mn and Fe changes by less than 0.5% when increasing the size of the cluster from 15 lattice sites including four Nb atoms and ten empty spheres in the 2NN shell to 52 lattice sites including the first five neighbor shells around the adatom. The anisotropy energy of the adatom between out-of-plane and in-plane orientations also changes by less than 4% between the two cluster sizes. The local density of states (LDOS) was calculated using 68 meV imaginary part of the complex energy in our KKR calculations, while moving along the real energy axis. We projected the LDOS onto real spherical harmonics in the absence of SOC.

### 2.3. Spin Model

Relying on the adiabatic decoupling of the electronic and spin degrees of freedom and on the rigid-spin approximation [47], the thermodynamic potential of a magnetic system is characterized by a set of unit vectors, $\left\{\vec{e}\right\}=\left\{{\vec{e}}_{1},{\vec{e}}_{2},\cdots ,{\vec{e}}_{N}\right\}$, corresponding to the orientations of the local magnetic moments. The grand potential $\Omega \left(\{\vec{e}\}\right)$ then defines a classical spin Hamiltonian which can be used in numerical simulations. Instead of calculating the grand potential directly, a straightforward idea is to map it onto a generalized Heisenberg model of the form
(1)$$\Omega \left(\left\{\vec{e}\right\}\right)=\Omega_{0}+\sum_{i=1}^{N}{\vec{e}}_{i}{\underline{\underline{K}}}_{i}{\vec{e}}_{i}-\frac{1}{2}\sum_{\begin{array}{c} i,j=1\\ i\neq j \end{array}}^{N}{\vec{e}}_{i}{\underline{\underline{J}}}_{ij}{\vec{e}}_{j},$$
where $\Omega_{0}$ is a constant, ${\underline{\underline{K}}}_{i}$ are the second-order single-ion anisotropy matrices, and ${\underline{\underline{J}}}_{ij}$ are the tensorial exchange interactions, which can be decomposed into three parts:(2)$$\begin{array}{cc} {\underline{\underline{J}}}_{ij}= & J_{ij}^{I}\underline{\underline{I}}+{\underline{\underline{J}}}_{ij}^{S}+{\underline{\underline{J}}}_{ij}^{A}, \end{array}$$
where
(3)$$\begin{array}{cc} J_{ij}= & \frac{1}{3}\mathrm{Tr}\left({\underline{\underline{J}}}_{ij}\right) \end{array}$$
is the isotropic exchange interaction,
(4)$$\begin{array}{cc} {\underline{\underline{J}}}_{ij}^{S}= & \frac{1}{2}\left({\underline{\underline{J}}}_{ij}+{\underline{\underline{J}}}_{ij}^{T}\right)-J_{ij}\underline{\underline{I}}, \end{array}$$
with *T* denoting the transpose of a matrix, is the traceless symmetric part of the matrix which is known to contribute to the magnetic anisotropy of the system (two-ion anisotropy), and the antisymmetric part of the matrix,
(5)$$\begin{array}{cc} {\underline{\underline{J}}}_{ij}^{A}= & \frac{1}{2}\left({\underline{\underline{J}}}_{ij}-{\underline{\underline{J}}}_{ij}^{T}\right) \end{array}$$
is related to the Dzyaloshinskii–Moriya (DM) interaction,
(6)$$\begin{array}{c} {\vec{e}}_{i}{\underline{\underline{J}}}_{ij}^{A}{\vec{e}}_{j}={\vec{D}}_{ij}\left({\vec{e}}_{i}\times {\vec{e}}_{j}\right) \end{array}$$
with the DM vector, $D_{ij}^{\alpha }=\frac{1}{2}\varepsilon_{\alpha \beta \gamma }J_{ij}^{\beta \gamma }$, $\varepsilon_{\alpha \beta \gamma }$ being the Levi–Civita symbol.

The site-resolved effective anisotropy matrix, including single-ion and two-ion contributions, can be defined as [25]
(7)$${\underline{\underline{A}}}_{i,\mathrm{FM}/\mathrm{AFM}}={\underline{\underline{K}}}_{i}-\frac{1}{2}\sum_{j=1}^{N}{\underline{\underline{J}}}_{ij}^{S}{\left(\pm 1\right)}^{i+j},$$
where the sign of (+1) and (−1) has to be used for dimers and chains with (NN) ferromagnetic (FM) and antiferromagnetic (AFM) couplings, respectively.

The ground state of the magnetic clusters is determined by subsequent low-temperature Metropolis Monte Carlo (MC) and zero-temperature Landau–Lifshitz–Gilbert (LLG) spin dynamics simulations. This procedure is especially important for Fe chains along the [$1\bar{1}0$] crystallographic direction to avoid local energy minima because they have spin spiral ground states with tiny energy difference between the states with opposite chiralities, as discussed in Section 3.3. The details of the MC simulations can be found in [43]. The accuracy of the ground state can be improved by the LLG spin dynamics simulations containing the damping term only, and starting from the final state of the MC simulations. For each system, ten runs with random initial configurations were performed, where we assumed that the actual ground state has been found if at least eight out of the ten runs resulted in the same final state with the lowest energy.

## 3. Results and Discussion

### 3.1. Mn and Fe Adatom

The SCF calculations for the Mn and Fe adatoms embedded into the first vacuum layer were performed containing atomic positions inside the radius of 331 pm (see Figure 1), which means beyond the adatom the cluster also contained four Nb atoms from the top Nb layer, and 10 vacuum positions.

First, let us consider the Mn adatom. The spin magnetic moment of the Mn atom is $3.70\;\mu_{B}$, close to the $3.60\;\mu_{B}$ value obtained from VASP (cf. Section 2.1). The induced spin moment of the NN Nb atom is $0.23\;\mu_{B}$, the 2NN Nb moment is again one order of magnitude smaller, which supports the assumption that farther atomic sites do not need to be included in the self-consistently treated cluster.

In the spirit of the magnetic force theorem, the magnetocrystalline anisotropy energy (MAE) was determined based on band energy differences between magnetic orientations. In order to avoid the ambiguity in the size and orientation of the induced moments, during the calculation of the MAE the initial local exchange field on the sites with induced magnetic moments was set to zero. We obtained 0.31 meV for the MAE between the [$1\bar{1}0$] (*x*, in-plane) and [110] (*z*, perpendicular) crystallographic directions. The MAE between the [001] (*y*, in-plane) and [110] (*z*) directions is 0.16 meV, meaning that the easy and medium directions are the *z* and *y* directions, respectively. Note that the band energy difference between the *x* and the *z* directions is 0.29 meV, and 0.16 meV between the *y* and the *z* directions if the induced moments are also considered, meaning that switching off the exchange field at sites with induced magnetic moments causes less than 7% error. The main benefit of doing so is that we can easily compare the magnetic energy of complex magnetic structures.

The LDOS on the Mn adatom is shown in Figure 2a, where the LDOS is projected onto the real spherical harmonics of the *d* orbitals. To visualize both spin channels, the LDOS of the minority spin channels is multiplied by −1. It can be seen that the in-plane *d* orbitals, $d_{xy}$ and $d_{x^{2}-y^{2}}$, typically display sharper peaks than the other *d* orbitals, because they hybridize less with the surface Nb atoms. Note that due to the $C_{2v}$ symmetry of the system, the $d_{z^{2}}$ and $d_{x^{2}-y^{2}}$ orbitals are hybridized with each other, displaying peaks at the same energies. The peak in the $d_{yz}$ channel is located at the lowest energy, which can be explained by the fact that the closest two Nb atoms are located in the *y*-*z* plane, and with the hybridization these states may gain the most energy. The large hybridization of this orbital with the substrate can be seen in the broad LDOS profile in the minority spin channel.

The Fe adatom was also calculated self-consistently with similar conditions as the Mn adatom, except for the different perpendicular relaxation described in Section 2.1. The spin magnetic moments of the Fe atom and of the NN Nb atom are $2.34\;\mu_{B}$ ($2.14\;\mu_{B}$ from VASP) and $0.22\;\mu_{B}$, respectively, while the other induced moments are less then $0.02\;\mu_{B}$. The MAEs are equal to $A^{xx}-A^{zz}=0.24$ meV and $A^{yy}-A^{zz}=0.70$ meV, meaning that *z* is the easy and *x* is the medium direction. The LDOS of the Fe adatom is shown in Figure 2b, where the more structured profiles indicate that most of the orbitals hybridize with the substrate more than in the case of the Mn adatom, which can be well understood by the larger relaxation. The Fe atom has one more electron than the Mn atom nominally possessing a half-filled *d* band, which leads to a higher occupation of the minority spin channel in the Fe adatom, indicated by a shift of the minority spin orbitals in the LDOS towards smaller energies. This also decreases the spin moment of Fe. Overall, the LDOS of the Fe adatom in Figure 2b seems to be in a good agreement with that reported in [8] without orbital decomposition.

### 3.2. Mn and Fe Dimers

We calculated magnetic dimers on the Nb(110) surface with different distances between the magnetic atoms and positioned along various crystallographic directions, with the *x*, *y* and *u* directions denoting the [$1\bar{1}0$], [001] and [$1\bar{1}1$] crystallographic directions, respectively. The dimers are labeled by a letter and a number together (α-*d*NN), where $\alpha \in \left\{x,y,u\right\}$, and *d*NN labels the distance of the two adatoms (see Figure 1 for an illustration), meaning the *d*-th-nearest neighbor. The environment was fabricated by the following procedure: first a chain with d+1 atoms was created with a 2NN environment, then the magnetic atoms were inserted to both ends of the d+1-atom-long chain; finally, Nb and vacuum atoms were inserted to the proper positions in the cluster. Thus, for example, in a *x*-2NN dimer, the atomic position between the two magnetic atoms, where a vacuum sphere was embedded, and its 2NN environment have also been included in the cluster (see Figure 1). During the SCF iterations, we used ferromagnetic (FM) ordering of the spins along *z*, assumed to be the easy direction; while the induced moments were relaxed.

The spin model parameters for the Mn and Fe dimers are collected in Table 1. In both cases, the *u*-1NN dimer has the largest isotropic interaction $J_{12}^{I}$ in absolute value. The main tendency of the isotropic interactions is that they decrease as the distance between the magnetic atoms is increased. Due to the $C_{2}$ symmetry of the dimers, the *z* component of the DM interaction $D_{12}^{z}$ vanishes. For the dimers along the *x* and *y* directions, the DM vector component perpendicular to the mirror plane that exchanges the two magnetic atoms also has to be 0. The MAE is similar as for the adatom, so the easy axis is close to the *z* direction, while the medium axis points nearly along the *y* (*x*) direction for the Mn (Fe) dimers, respectively. The anisotropy matrices *A* have finite off-diagonal components, which are not explicitly listed in Table 1 but indicated by the deviation of the dimers from the collinear alignment, discussed below. This is because the $C_{2}$ symmetry of the *u*-1NN, *u*-2NN, and *u*-3NN dimers does not determine the anisotropy directions to be parallel to the principal Cartesian axes. In the case of dimers along the *x* and *y* directions, the $C_{2v}$ symmetry determines one of the anisotropy axes to point along a Cartesian direction, namely, along *y* for the *x* dimers and along *x* for the *y* dimers, as a consequence of the mirror plane leaving the magnetic atoms invariant.

The ground state spin angle between the spin moments of the dimer atoms was determined by spin dynamics simulations, listed as $\vartheta_{\mathrm{SD}}$ in Table 1. Due to the hierarchy of the parameters, where the isotropic interaction is the largest, the ground state spin angle is close to $0^{\circ }$ (FM alignment) or $180^{\circ }$ (AFM alignment), determined by the sign of the isotropic interaction, where the positive (negative) sign corresponds to the FM (AFM) coupling. The DMI, preferring a perpendicular alignment of the spins, causes a deviation from the collinear alignment. The deviation of the easy anisotropy axis from the *z* direction discussed above contributes to this effect, as the easy direction is not parallel on the two adatoms, rather connected by a $C_{2}$ rotation. Although the ground-state angle cannot be expressed in a closed form using the interaction parameters, an approximate method of finding it is discussed in Appendix A. These values are given as $\vartheta_{AD}$ in the table, and agree with the numerically calculated values $\vartheta_{\mathrm{SD}}$ within $0.1^{\circ }$ accuracy. The AFM ground states of the Mn *u*-1NN and Mn *y*-2NN dimers agree well with scanning tunneling spectroscopy experiments [7], but the Mn *x*-1NN dimer exhibits experimentally a FM ground state contrary to the AFM state obtained from our calculations. This may be caused by the limitations of the simulation techniques, e.g., the angular momentum cut-off, or additional structural relaxations that are not taken into account in the KKR calculations.

For further analysis, in the non-relativistic case (without SOC), we calculated the orbital decomposition of the isotropic interaction for the closest (1NN) dimers along the different directions, $J_{12,\mathrm{nr}}^{I}$, which can be seen in Table 2. The main contributions come from the *d* orbitals. The $d_{z^{2}}$ orbital has a significant contribution in all dimers but the Fe *u*-1NN case, as it can hybridize with the underlying Nb surface. The in-plane components, $d_{xy}$ and $d_{x^{2}-y^{2}}$, are more relevant for the closer dimers, indicating a direct hybridization of these orbitals between the magnetic atoms. Note that the $d_{yz}$ orbital contributes to $J_{12,\mathrm{nr}}^{I}$ most strongly in the case of the *y*-1NN dimers, possibly mediated by the Nb atom located below the center of the magnetic atoms along that direction. Interestingly, the $d_{xz}$ orbital contributes most strongly in the case of the *u*-1NN dimers. Its contribution to $J_{12,\mathrm{nr}}^{I}$ in the *x*-1NN dimers is rather weak, which may be attributed to the large distance between the atoms along the *x* direction.

The orbital-decomposed isotropic exchange interactions are possible to trace back to the LDOS of the dimers, illustrated for the Mn *u*-1NN dimer in Figure 3a. To get more accurate results, we recalculated the SCF potentials with an AFM alignment of the spins, while the orientations of the induced moments were relaxed. The LDOS was also calculated in an AFM alignment. Compared with the Mn adatom in Figure 2a, the main LDOS features are the same, the positions of the peaks do not visibly shift, but due to the adjacency of the Mn atoms, the curves become flatter. Note that a FM ordering of the spins would cause more considerable changes in the LDOS, because in that case the same spin channel of the two Mn atoms in the same energy range could hybridize, while in the AFM case the majority spin channel of either atom and the minority spin channel of the other atom are shifted in energy, leading to much less hybridization.

The orbital decomposition of the LDOS can also be applied to the band energy. To have a deeper look on the role of the orbital contributions to the isotropic exchange interaction, we introduce the band energy difference between AFM and FM configurations as
(8)$$\Delta E_{\mathrm{band},D}^{\gamma }\left(E_{F}\right)=E_{\mathrm{band},D}^{\mathrm{AFM},\gamma }\left(E_{F}\right)-E_{\mathrm{band},D}^{\mathrm{FM},\gamma }\left(E_{F}\right),$$
where D indicates that it is obtained from the LDOS, and γ stands for the atomic orbital. The band energy difference can directly be calculated from the change of the LDOS,
(9)$$\Delta E_{\mathrm{band},D}^{\gamma }\left(E\right)=\int_{-\infty }^{E}d\varepsilon \left(\varepsilon -E_{F}\right)\left({\mathrm{LDOS}}^{\mathrm{AFM},\gamma }\left(\varepsilon \right)-{\mathrm{LDOS}}^{\mathrm{FM},\gamma }\left(\varepsilon \right)\right),$$
where $E_{F}$ is the Fermi energy. $\Delta E_{\mathrm{band},D}^{\gamma }$ as a function of energy is shown in Figure 3b, where we performed the energy integration parallel to the real energy axis, with 68 meV imaginary part of the energy. In the majority spin channel larger oscillations can be seen than for the minority spin channel, but its contribution averages out if the integral is evaluated up to the Fermi energy. It can be concluded that the magnetic orientation modifies the sharpness of the LDOS, but does not affect the positions of the peaks, so in the case of fully occupied states, only tiny contributions to the band energy difference can be observed. This is also true for the minority spin channel when the whole bandwidth is considered but at the Fermi energy those orbitals are only partially occupied, leading to relatively large band-energy differences.

Instead of calculating the band energy along the line parallel to the real energy axis, we integrated it using the same semicircle contour that was used in the SCF calculations to get more accurate results. $E_{\mathrm{band},D}^{\mathrm{AFM},\gamma }$ and $E_{\mathrm{band},D}^{\mathrm{FM},\gamma }$ were calculated using the original, FM SCF potentials, with the initial local exchange field set to zero on the sites with induced magnetic moments. The band energy differences for the whole cluster divided by the number of magnetic atoms is denoted by $\Delta E_{\mathrm{band},L}$ in Table 3, where the L subscript indicates that Lloyd’s formula has been applied instead of Equation (Equation 9). In terms of a spin model, the energy difference between the AFM and FM configurations is expected to be $\Delta E_{\mathrm{band}}=J$. Indeed, we find that $\Delta E_{\mathrm{band},L}$ agrees with the non-relativistic isotropic exchange coupling, $J_{12,\mathrm{nr}}^{I}$, in Table 2 up to a precision of 0.5 meV, which verifies the SCE spin model calculations. In most of the cases, a semiquantitative agreement can be concluded between the orbital-decomposed isotropic spin interactions (Table 2) and the orbital-decomposed band energies restricted to a single magnetic atom (Table 3), where the signs and the relative magnitudes of the decomposed parameters are the same. Note that the sum of the orbital contributions does not equal $\Delta E_{\mathrm{band},L}$, mainly because the latter also includes band energy differences on the neighboring Nb atoms in the cluster. The closest agreement between the sum and the total band energy difference is found for the *u*-1NN dimers, because in this case the magnetic atoms are closer to each other and the direct scattering between the magnetic atoms is more relevant than that mediated by the Nb atoms.

### 3.3. Mn and Fe Chains

In the following, monatomic NN-distanced Mn and Fe chains containing 5, 10, and 15 magnetic atoms along the *x*, *y*, and *u* directions are considered. Similarly as for the dimers, a second-neighbor environment was included in the self-consistently treated clusters, see Figure 1 for the y5 chain as an example. The spin model parameters for the chains were determined, and the parameters at the end (edge atom) and in the middle (center atom) of the 15-atom-long chains are reported in Table 4.

The NN isotropic interactions can directly be compared with the values obtained for the NN dimers. The interactions at the ends of the chains agree within 8% with the dimer values for the Mn systems, but vary much more for the Fe systems. The NN isotropic interaction is homogeneous along the Mn chains, e.g., 29.09 and 27.14 meV for the Mn y15 chain at the end and in the middle of the chain, respectively. This value is much more sensitive to the coordination number in the case of the Fe chains, where the same quantities are 25.42 and 18.57 meV for the Fe y15 chain. In order to explain why the isotropic exchange interaction varies more strongly from the dimer through the chain’s end to the middle of the chain in Fe systems compared to Mn ones, we calculated the LDOS at sites 1 and 8 of the Fe y15 chain, shown in Figure 4. For the edge atom, the order of the peaks and their shape is similar to the LDOS of the adatom in Figure 2b, but the widths of the peaks differ: the $d_{xy}$ and $d_{x^{2}-y^{2}}$ peaks in the majority spin channel become wider, but the $d_{yz}$ peak in the majority channel and the $d_{xy}$ peak in the minority channel are sharper. Note that the LDOS of the $d_{x^{2}-y^{2}}$ orbital in the minority spin channel splits into two peaks. All the changes become stronger as we move toward the middle of the chain, the sharper peaks get even sharper, the splitting of the minority $d_{x^{2}-y^{2}}$ peak is larger too, and now the majority $d_{xy}$ peak also splits. Despite the increased number of neighbors of the center atom, the sharper features in the LDOS may be attributed to the fact that the center atom occupies a position with $C_{2v}$ symmetry where only the $d_{z^{2}}$ and the $d_{x^{2}-y^{2}}$ orbitals hybridize. At the end of the chain, only a mirror symmetry on the *y*-*z* plane is preserved, leading to a hybridization between the $d_{z^{2}}$, $d_{x^{2}-y^{2}}$, $d_{yz}$ and $d_{xy}$, $d_{xz}$ orbitals, respectively. Because of the higher filling of the minority band of Fe compared to Mn, the small changes in the shapes of the different orbitals below the Fermi level are expected to have a more pronounced effect on the magnetic interactions, similar to what was demonstrated for a dimer in Figure 3b. To characterize the influence of the atomic environment, we calculated the band energy differences between AFM and FM magnetic configurations similarly to the case of the dimers, $\Delta E_{\mathrm{band},D}^{i}$, where *i* denotes the atomic site in the chain. When compared to the atomistic spin model, it is expected that $\Delta E_{\mathrm{band},D}^{i}$ is approximately equal to the sum of the NN isotropic exchange interactions of site *i*, $J_{i\left(i-1\right)}^{I}+J_{i\left(i+1\right)}^{I}$, as these parameters have by far the largest magnitude in Table 4. This approximation is further supported by the fact that $J_{i(i+2)}^{I}$ does not contribute to $\Delta E_{\mathrm{band},D}^{i}$, as the 2NNs are parallel both in the FM and in the AFM configuration, and the interactions with farther neighbors are even weaker. $\Delta E_{\mathrm{band},D}^{1}$ can directly be compared with the isotropic coupling, because the first site has a single NN only, but needs to be divided by 2 for the center atom, where $J_{87}^{I}=J_{89}^{I}$. We obtained $\Delta E_{\mathrm{band},D}^{1}=21.04\;\mathrm{meV}$ and $\Delta E_{\mathrm{band},D}^{8}/2=16.13\;\mathrm{meV}$ for the Fe y15 chain, while $\Delta E_{\mathrm{band},D}^{1}=33.15\;\mathrm{meV}$, $\Delta E_{\mathrm{band},D}^{8}/2=20.07\;\mathrm{meV}$ for the Fe u15 chain. These band energy differences divided by the coordination number are very close to the corresponding $J_{i\left(i+1\right)}^{I}$ values in Table 4. This way, we validated the inhomogeneity of the NN isotropic exchange interactions in the Fe chains by two examples, because these features are also present in the change of the LDOS.

Carrying on with the discussion of Table 4, due to the $C_{2v}$ symmetry of the chains along *x* and *y*, the DM vector components have to vanish in the mirror plane that leaves the position of the magnetic atoms unchanged, corresponding to the *z* component along both directions and the *x* and the *y* components for chains along the *x* and *y* directions, respectively. For chains along the *u* direction with $C_{2}$ symmetry, the *z* component of the DM vector between sites connected by the $180^{\circ }$ rotation also has to be zero. $D_{i\left(i+1\right)}^{z}$ takes a finite value on the order of 0.1 meV at the ends of the u15 chains, which is still weaker than the other spin interaction parameters listed in Table 4 for these pairs. The DM interactions are stronger if the atoms are located closer to each other in the chains, but overall they are relatively weak compared to the isotropic exchange interactions.

The values of the anisotropy tensor elements at the ends of the chains are similar to those obtained for the NN dimers, e.g., $A^{xx}-A^{zz}=0.29\;\mathrm{meV}$ for the Mn *x*-1NN dimer and 0.30meV for the first site (edge atom) in the Mn x15 chain. This difference is below 7% for all the anisotropy parameters, so one can conclude that the anisotropy is mostly determined by the local environment, which is similar at the end of NN chains and in the case of the NN dimers. Moving toward the middle of the chain, the anisotropy of Mn chains, interestingly, is more sensitive to the coordination number than that of Fe chains, in the opposite manner that was observed for the isotropic couplings.

The ground state of all the Mn chains and of the Fe chains along *y* and *u* directions is determined by the sign of the NN isotropic interaction: if it is positive (negative), then the ground state is the FM (alternating AFM) configuration. The spins point along ±z, with some deviation at the ends of the chains due to the DM interaction, and due to the small changes in the easy direction along the chains, based on similar symmetry arguments to how the ground-state angle in the dimers was determined in Section 3.2. Based on scanning tunneling spectroscopy measurements, FM and AFM ground states were found for Mn *u*13 and for Mn *y*15 chains, respectively [28], in agreement with our results.

We obtain a spin spiral ground state for the Fe chains along the *x* direction, see Figure 5. This ground state can be well understood based on the Fourier transform of the spin model parameters of the x15 chain shown in Figure 6,
(10)$$J^{k}\left(q\right)=\sum_{j=1}^{k}J_{8\left(8+j\right)}^{I}\cos\left(jaq\right),\;q\in \left[-\frac{\pi }{a_{x}},\frac{\pi }{a_{x}}\right],$$
and
(11)$$D^{k}\left(q\right)=\sum_{j=1}^{k}D_{8\left(8+j\right)}^{y}\sin\left(jaq\right),\;q\in \left[-\frac{\pi }{a_{x}},\frac{\pi }{a_{x}}\right],$$
where $a_{x}=\sqrt{2}a_{\mathrm{Nb}}=466.74\;\mathrm{pm}$. These formulae describe a harmonic spin spiral state rotating in the *x*-*z* plane, as the DM interaction only contributes to the energy of spin spirals in this plane. k=1 in Figure 6 represents a simple harmonic. Due to the frustration of the 2NN isotropic interaction which is also AFM, starting from k=2 we find local minima at positive and negative wave numbers, corresponding to opposite chiralities. If at least four shells are taken into account in the summation, we obtain a flat dispersion relation, which, together with the anisotropy, can stabilize spin spirals with several different wave vectors. While the DM interaction prefers a clockwise rotation of the spins (see Figure 5), we confirmed by Landau–Lifshitz–Gilbert dynamics simulations that the relatively large anisotropy also stabilizes the opposite rotational sense as a metastable state. This suggests that in very long chains, sections with different spin spiral periods and even opposite chiralities may alternate at finite temperature if the system is unable to find the global energy minimum. Such a state would be similar to the spin glass state recently investigated on the (0001) surface of Nd in [27]. If all the interactions are taken into account up to the 7th neighbor of the middle spin, the maximum of $J^{k}\left(q\right)+D^{k}\left(q\right)$ is obtained at $q_{\max}=0.59\;\pi /a_{x}$, meaning that the wavelength of the spin spiral should be $3.39\;a_{x}$. The ground state spin angle between sites 8 and 9 is $107^{\circ }$ in the simulations, which translates to a spin spiral wavelength of $3.35\;a_{x}$, being in excellent agreement with the value determined from the Fourier transform.

## 4. Summary

Motivated by recent interest in so-called single chain magnets as well as in magnetic adatoms, dimers and atomic chains on superconducting surfaces, we performed first-principles calculations for Mn and Fe clusters on Nb(110) in order to determine their electronic and magnetic properties. We determined the magnetic interactions using the spin cluster expansion and compared the isotropic exchange interactions to band energy differences between the ferromagnetic and antiferromagnetic configurations. Based on the orbital decomposition of the isotropic exchange interaction, we concluded that the in-plane $d_{xy}$ and $d_{x^{2}-y^{2}}$ orbitals, describing direct exchange, have the strongest relative contribution if the magnetic atoms are closely packed along the *u* direction. The $d_{z^{2}}$ orbital mediating the exchange through the substrate Nb atoms has a significant contribution for dimers along all considered crystallographic directions, while the $d_{yz}$ orbital influences the isotropic exchange stronger if the dimer is oriented along the lobes of this orbital. These contributions can be quantitatively traced back to the different hybridization of the atomic orbitals between ferromagnetic and antiferromagnetic spin configurations observable in the LDOS. The fully occupied majority spin channels have been demonstrated to contribute less to the magnetic interactions than the partially occupied minority spin channels in the dimers. The higher occupation of the minority band in Fe compared to Mn leads to a larger variation of the NN isotropic exchange interaction with the local environment, i.e., by going from a dimer through the end of a chain to the middle of a chain along the same crystallographic direction.

The ground state of the dimers is mainly determined by the sign of the isotropic interaction, where other spin model parameters, namely, the DM interaction and the tilting of the easy direction, only slightly perturb the FM or AFM alignment of the spins in the dimers. The same argument holds for the Mn and Fe chains along *u* and *y* directions, where the ground state is the FM or the alternating AFM configuration if the NN isotropic interaction is FM or AFM, respectively, with slight deviations at the ends of the chains. This was attributed to the relatively weak SOC in the Nb substrate. The Fe chains along the *x* = [$1\bar{1}0$] direction display a spin spiral ground state caused by the frustration of the NN and 2NN AFM isotropic couplings, and the wavelength of the spin spiral in the Fe x15 chain is quantitatively well reproduced by the Fourier transform of the isotropic and DM interactions. A flat spin spiral dispersion relation is identified in this chain, which, together with the magnetic anisotropy, can stabilize spin spirals with various wave vectors and chiralities. Due to this, segments with different periods and chiralities may coexist in longer chains.

## Figures and Tables

**Figure 1 nanomaterials-11-01933-f001:**
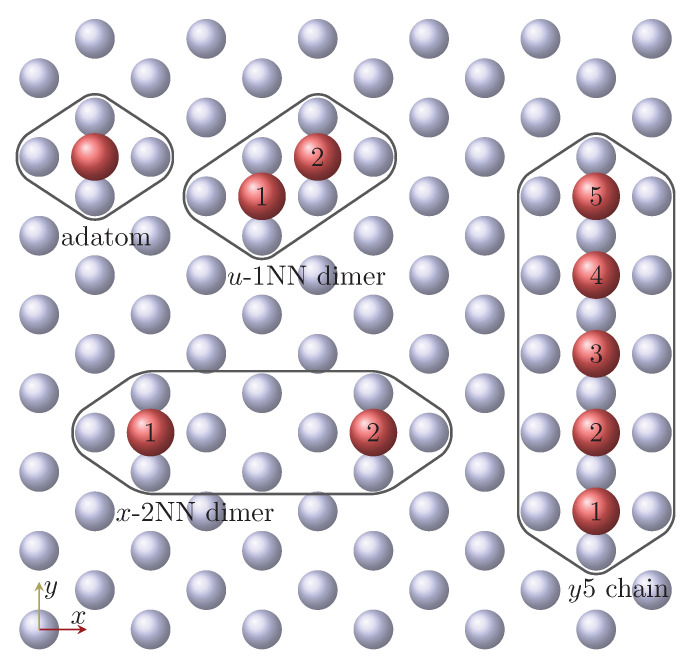
Illustration of the atomic clusters. Red balls represent the magnetic atoms at hollow positions on the bcc(110) surface, and gray balls represent the top Nb atoms. The clusters were calculated one by one, and contained the atoms inside the black curves. Crystallographic directions: *x* = [$1\bar{1}0$], *y* = [001], *u* = [$1\bar{1}1$].

**Figure 2 nanomaterials-11-01933-f002:**
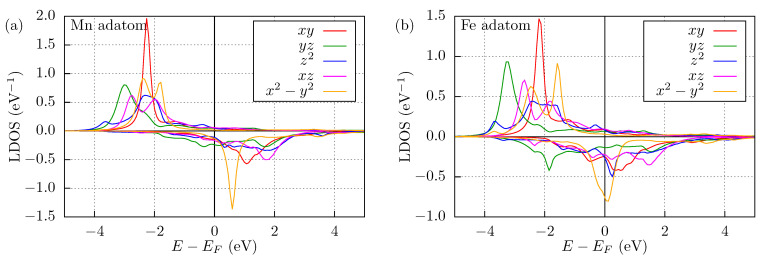
Local density of states of magnetic adatoms. Spin-resolved LDOS of the (**a**) Mn (cf. the work in [7]) and of the (**b**) Fe adatoms as projected onto the *d* orbitals. Positive values: LDOS of majority spin channel; negative values: LDOS of minority spin channel multiplied by −1. The Fermi energy is denoted by a vertical black line at E=0.

**Figure 3 nanomaterials-11-01933-f003:**
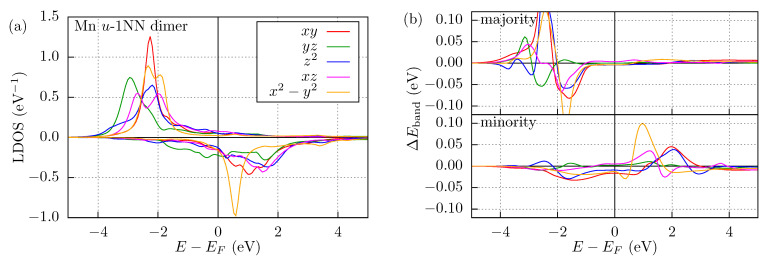
Local density of states and band energy differences of the Mn *u*-1NN dimer. (**a**) Spin-resolved LDOS in an AFM configuration as projected onto the *d* orbitals. (**b**) Spin-resolved band energy differences between AFM and FM alignments as a function of energy according to Equation (Equation 9). The Fermi energy is denoted by a vertical black line at E=0.

**Figure 4 nanomaterials-11-01933-f004:**
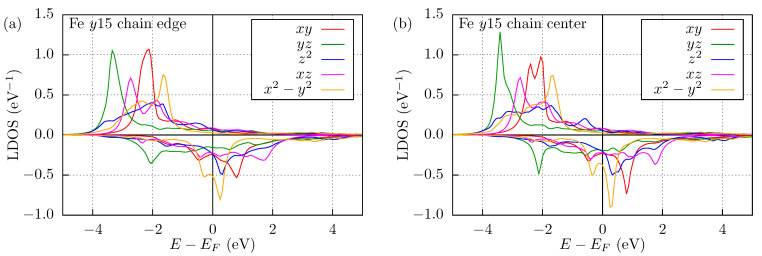
Local density of states of the Fe y15 chain. Spin-resolved LDOS of (**a**) the edge and (**b**) the center atom. Positive and negative values correspond to majority and minority spin channels, respectively. The Fermi energy is denoted by a vertical black line at E=0.

**Figure 5 nanomaterials-11-01933-f005:**
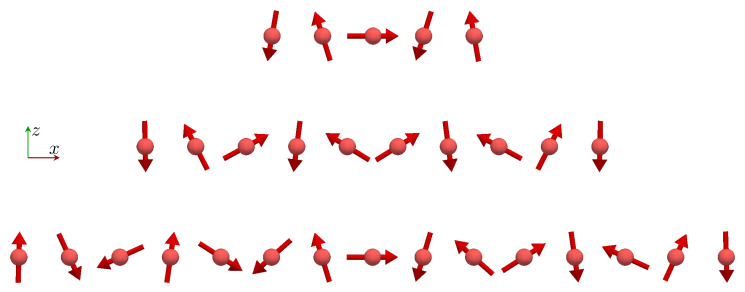
Side view of the ground state of the Fe chains along the *x* direction. The spin configurations of the Fe x5, x10, and x15 chains are shown. With increasing *x* coordinate, a clockwise rotation of the spins is observed for all chain lengths.

**Figure 6 nanomaterials-11-01933-f006:**
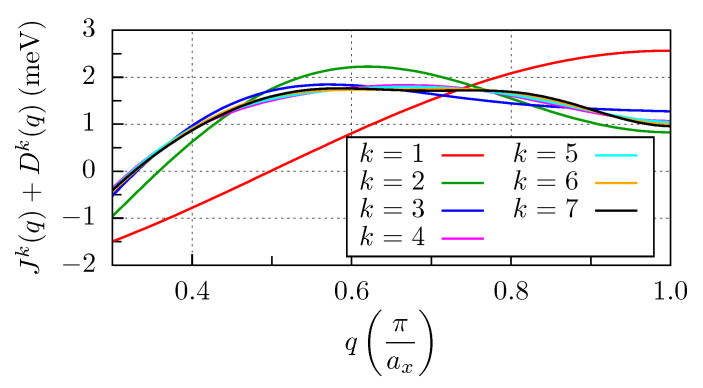
Fourier transform of the magnetic interactions for the Fe x15 chain. The isotropic and the DM interactions in Equations (Equation 10) and (Equation 11) are summed up.

**Table 1 nanomaterials-11-01933-t001:** Spin model parameters in units of meV and ground state spin angles in degrees for the Mn and Fe dimers. The isotropic interactions $J_{ij}^{I}$ (positive and negative values mean FM and AFM coupling, respectively), DM interaction vectors ${\vec{D}}_{ij}$ and anisotropy matrix elements $\underline{\underline{A}}$ are shown. Note that $\underline{\underline{A}}$ also includes the two-site anisotropy. $D_{12}^{z}=0$ holds for all the dimers. $\vartheta_{\mathrm{SD}}$ is the ground state angle based on spin dynamics simulations, and $\vartheta_{AD}$ is the approximate ground state spin angle calculated using the spin model parameters (see Appendix A). The $\vartheta_{\mathrm{SD}}$ angles of the Mn *u*-1NN, Mn *x*-1NN, and Mn *y*-2NN dimers were reported in  [7].

	$J_{12}^{I}$	$D_{12}^{x}$	$D_{12}^{y}$	$A^{\mathrm{xx}}-A^{\mathrm{zz}}$	$A^{\mathrm{yy}}-A^{\mathrm{zz}}$	$\vartheta_{\mathrm{SD}}$	$\vartheta_{\mathrm{AD}}$
Mn *x*-1NN	−7.13	0.00	0.01	0.29	0.16	179.93	179.90
Mn *x*-2NN	−1.90	0.00	−0.02	0.28	0.14	179.04	179.04
Mn *x*-3NN	−0.45	0.00	0.03	0.28	0.14	178.16	178.18
Mn *y*-1NN	31.93	0.22	0.00	0.38	0.18	0.36	0.37
Mn *y*-2NN	−1.04	−0.17	0.00	0.28	0.16	172.24	172.13
Mn *y*-3NN	−0.10	−0.01	0.00	0.28	0.14	178.04	178.05
Mn *u*-1NN	−33.00	0.10	0.32	0.34	0.24	179.46	179.46
Mn *u*-2NN	5.92	0.08	−0.55	0.28	0.12	5.24	5.27
Mn *u*-3NN	−1.32	0.15	−0.08	0.28	0.12	172.90	172.88
Fe *x*-1NN	−4.35	0.00	−0.15	0.32	0.70	178.04	178.04
Fe *x*-2NN	−2.92	0.00	−0.06	0.29	0.67	178.93	178.93
Fe *x*-3NN	−0.74	0.00	0.11	0.28	0.68	173.94	173.93
Fe *y*-1NN	33.30	0.07	0.00	0.29	0.73	0.05	0.07
Fe *y*-2NN	10.61	0.02	0.00	0.27	0.68	0.01	0.02
Fe *y*-3NN	−0.73	0.14	0.00	0.29	0.68	174.67	174.72
Fe *u*-1NN	49.66	1.39	−3.24	0.45	0.70	4.00	4.01
Fe *u*-2NN	9.85	0.00	−0.85	0.36	0.77	4.95	4.98
Fe *u*-3NN	−3.91	0.15	−0.03	0.29	0.68	177.60	177.59

**Table 2 nanomaterials-11-01933-t002:** Orbital decomposition of the isotropic exchange interaction in the 1NN Mn and Fe dimers. All values are given in meV units. Note that the SOC is turned off so the $J_{12,\mathrm{nr}}^{I}$ values slightly differ from the $J_{12}^{I}$ parameters given in Table 1.

	$J_{12,\mathrm{nr}}^{I}$	*s*	*p*	$d_{\mathrm{xy}}$	$d_{\mathrm{yz}}$	$d_{z^{2}}$	$d_{\mathrm{xz}}$	$d_{x^{2}-y^{2}}$	*f*
Mn *x*-1NN	−7.27	−0.12	0.26	0.45	0.55	−7.18	−0.38	−0.87	0.02
Mn *y*-1NN	31.75	0.16	−0.55	2.47	18.66	12.15	0.87	−1.99	−0.02
Mn *u*-1NN	−33.19	−2.75	3.18	−12.08	2.22	−14.16	15.71	−25.76	0.45
Fe *x*-1NN	−4.59	−0.05	0.11	1.41	1.47	−7.05	−0.70	0.20	0.01
Fe *y*-1NN	33.53	0.59	−0.65	3.17	5.26	6.06	−1.53	20.63	−0.01
Fe *u*-1NN	50.01	−0.26	0.12	16.82	1.44	0.33	15.74	15.81	0.02

**Table 3 nanomaterials-11-01933-t003:** Band energy differences between AFM and FM configurations in the 1NN Mn and Fe dimers. The band energy difference calculated for a single magnetic atom in the cluster is decomposed into atomic *d* orbitals according to Equation (Equation 9), and the majority and minority components are summed. The band energy differences of the whole self-consistently treated clusters between AFM and FM spin configurations divided by the number of magnetic atoms are also given based on Lloyd’s formula ($\Delta E_{\mathrm{band},L}$). All reported values are in meV units.

	$\Delta E_{\mathrm{band},D}^{\mathrm{xy}}$	$\Delta E_{\mathrm{band},D}^{\mathrm{yz}}$	$\Delta E_{\mathrm{band},D}^{z^{2}}$	$\Delta E_{\mathrm{band},D}^{\mathrm{xz}}$	$\Delta E_{\mathrm{band},D}^{x^{2}-y^{2}}$	$\Delta E_{\mathrm{band},L}$
Mn *x*-1NN	0.19	0.46	−2.13	0.07	−0.32	−7.13
Mn *y*-1NN	3.24	11.40	7.32	0.26	−0.39	31.99
Mn *u*-1NN	−20.96	2.84	−13.95	10.56	−18.50	−33.06
Fe *x*-1NN	1.08	0.65	−2.20	0.42	0.66	−4.13
Fe *y*-1NN	5.39	−1.37	4.51	−0.84	17.59	33.74
Fe *u*-1NN	10.78	1.35	0.50	13.80	21.18	49.94

**Table 4 nanomaterials-11-01933-t004:** Spin model parameters of the 15-atom-long chains in meV units. i=1 and 8 denote the edge atom (at the end) and center atom (in the middle) of the chains, respectively. For the notation of the spin model parameters see Table 1. Here, the 2NN isotropic exchange interactions, $J_{i(i+2)}^{I}$, are also reported.

	*i*	$J_{i(i+1)}^{I}$	$D_{i(i+1)}^{x}$	$D_{i(i+1)}^{y}$	$J_{i(i+2)}^{I}$	$A_{i}^{\mathrm{xx}}-A_{i}^{\mathrm{zz}}$	$A_{i}^{\mathrm{yy}}-A_{i}^{\mathrm{zz}}$
Mn x15	1	−6.87	−0.00	0.07	−1.28	0.31	0.16
Mn x15	8	−6.70	0.00	0.11	−1.15	0.34	0.18
Mn y15	1	29.09	0.24	0.00	2.31	0.41	0.23
Mn y15	8	27.14	0.30	0.00	2.14	0.57	0.32
Mn u15	1	−35.56	0.38	−0.55	2.71	0.39	0.26
Mn u15	8	−38.15	0.45	−1.50	2.58	0.49	0.40
Fe x15	1	−3.41	−0.00	−0.04	−2.02	0.32	0.71
Fe x15	8	−2.56	−0.00	0.01	−1.74	0.36	0.76
Fe y15	1	25.42	−0.19	0.00	10.00	0.23	0.73
Fe y15	8	18.57	−0.41	−0.00	9.64	0.18	0.77
Fe u15	1	34.11	0.82	−2.52	1.54	0.45	0.74
Fe u15	8	18.00	0.53	−1.68	3.80	0.51	0.72

## Data Availability

The data presented in this study are available on a reasonable request from the corresponding author.

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
