# Peer review of "Electronic and Magnetic Properties of Building Blocks of Mn and Fe Atomic Chains on Nb(110)"

_nanomaterials, 2021, doi:10.3390/nano11081933_

Round 1
Reviewer 1 Report
This work is a systematic and comprehensive theoretical study on the electronic and magnetic properties (i.e., ground-state, FM/AFM configurations, magnetic moments, isotropic interactions, etc.) of Mn and Fe clusters (namely, atomic, dimers and linear chains) on the Nb (110) surface.
The theoretical methodology is suitable and consistent with the purposes of the work. Additionally, authors developed a spin model to derive some useful parameters that allow determining the perturbation of the atomic clusters on the original FM/AFM alignments of the NB substrate.
Authors present a set of interesting results, which can uniquely be obtained by means of computational chemistry tools, so that they are of high relevance for the magnetic nanomaterials community and related applications (magnetic superconducting-based devices, quantum computers, etc.).
Conclusions are fully consistent with the results obtained and the derived messages.
Thus, I recommend the paper almost as it stands. I just encourage authors to take my points below (one related to the methods, the other to the surface model) into consideration:
- Is it not necessary to use the U term (i.e., PBE+U) for these calculations? Since I guess it isn’t, it would be fair, at least, to justify the use of pure PBE with any reference that demonstrates that PBE is good enough to describe the electronic and magnetic structure of this kind of systems.
- Why is so important to use such large supercell size for the modelled systems?
Author Response
Reviewer 1:
This work is a systematic and comprehensive theoretical study on the electronic and magnetic properties (i.e., ground-state, FM/AFM configurations, magnetic moments, isotropic interactions, etc.) of Mn and Fe clusters (namely, atomic, dimers and linear chains) on the Nb (110) surface.
The theoretical methodology is suitable and consistent with the purposes of the work. Additionally, authors developed a spin model to derive some useful parameters that allow determining the perturbation of the atomic clusters on the original FM/AFM alignments of the NB substrate.
Authors present a set of interesting results, which can uniquely be obtained by means of computational chemistry tools, so that they are of high relevance for the magnetic nanomaterials community and related applications (magnetic superconducting-based devices, quantum computers, etc.).
Conclusions are fully consistent with the results obtained and the derived messages.
Thus, I recommend the paper almost as it stands. I just encourage authors to take my points below (one related to the methods, the other to the surface model) into consideration:
Is it not necessary to use the U term (i.e., PBE+U) for these calculations? Since I guess it isn’t, it would be fair, at least, to justify the use of pure PBE with any reference that demonstrates that PBE is good enough to describe the electronic and magnetic structure of this kind of systems.
Authors' reply:
The electronic structure of the considered and similar metallic systems can usually be well described by using simple LDA and GGA density functionals. There is numerous evidence in the literature from many groups, most notably the Blügel group in Jülich, Germany (e.g. LDA in [9] and [10]), the Eriksson group in Uppsala, Sweden (e.g. LDA in [26] and [27]), also our group in Budapest (see references in the paper), etc. Ref. [8] explicitly used PBE for the same system. The usage of DFT+U is needed when such simple functionals fail to describe the electronic structure properly, for example systems having band gaps, like semiconductors, insulators, or where strong electron-electron interactions are present.
We included a comment on this in Section 2 in the revised version of the manuscript.
Reviewer 1:
Why is so important to use such large supercell size for the modelled systems?
Authors' reply:
For determining the structural relaxation of the magnetic adatoms and their vicinity, a 7x7 surface cell was used within the plane-wave method VASP. The reason is that with this size of supercell any artificial (non-physical) interactions can be excluded between the magnetic adatoms in repeating supercells since the closest distance between them is set to 2 nm.
We extended the description in Section 2.1. accordingly.
Reviewer 2 Report
The authors study the magnetic configurations of Mn and Fe clusters and chains on top of Nb surfaces. They perform ab-initio simulations and a spin model to characterize the magnetic interactions and their influence on the possible magnetic configurations. They find, among other things, that the results depend on the crystallographic direction and coordination number and that Fe chains develop a spin-spiral configuration.
I find the study scientifically sound, interesting and rigorous. The article is well written, the figures and tables are clear and the formulas and physical derivations seem to be correct. I think it can be further considered after the authors address the following concerns:
- It is not clear why they use two types of calculations (VASP code and KKR) and why in one case they use the GGA and in the other the LDA as exchange-correlation. This seems to be inconsistent, apart from the fact that LDA does not seem to work very well for this type of magnetic systems.
- The number of layers of Nb seems to be rather small (at least in the VASP calculation) and it is probably not converged. The authors should explain why they used such numbers and if the results are properly converged.
- The argument that the superconducting state might not affect the magnetic patterns is not convincing and it does not imply that the same results would be valid on a superconducting surface, where the electronic structure and interaction between the surface and atoms might be different. Can the authors elaborate on that?
- Although in general the physical explanations are correct, there is in some cases a lack of physical insight and the authors simply describe what they find (e.g. values of the anisotropy tensor in page 10). The authors should delve more into the physics in general.
Author Response
Reviewer 2:
The authors study the magnetic configurations of Mn and Fe clusters and chains on top of Nb surfaces. They perform ab-initio simulations and a spin model to characterize the magnetic interactions and their influence on the possible magnetic configurations. They find, among other things, that the results depend on the crystallographic direction and coordination number and that Fe chains develop a spin-spiral configuration.
I find the study scientifically sound, interesting and rigorous. The article is well written, the figures and tables are clear and the formulas and physical derivations seem to be correct. I think it can be further considered after the authors address the following concerns:
- It is not clear why they use two types of calculations (VASP code and KKR) and why in one case they use the GGA and in the other the LDA as exchange-correlation. This seems to be inconsistent, apart from the fact that LDA does not seem to work very well for this type of magnetic systems.
Authors' reply:
Since the KKR method does not calculate forces acting on atoms, and atomic geometry optimization is missing there, the VASP code has been solely used to determine the equilibrium atomic geometry for the magnetic adatoms on the Nb(110) substrate to be further used within KKR calculations. This is explicitly stated in the beginning of Section 2.1: "The lowest-energy atomic geometries of the Mn and Fe adatoms on the Nb(110) surface were determined separately by using the Vienna Ab-initio Simulation Package (VASP)", and at the end of Section 2.1: "These atomic geometries were used in the subsequent KKR calculations..." This is a conventional approach to combine VASP and KKR methods in this way, see e.g. numerous works of the Ebert group in Munich, Germany, also our group in Budapest (see references in the paper), among others.
The electronic structure of the considered and similar metallic systems can usually be well described by using simple LDA and GGA density functionals. There is numerous evidence in the literature from many groups, most notably the Blügel group in Jülich, Germany (e.g. LDA in [9] and [10]), the Eriksson group in Uppsala, Sweden (e.g. LDA in [26] and [27]), also our group in Budapest (see references in the paper), etc. Ref. [8] explicitly used PBE for the same system. We chose the GGA-PBE for the VASP calculations to be able to directly compare our results with Ref. [8]. However, the equilibrium atomic geometry should be practically insensitive to the GGA or LDA functional for such metallic systems, so we expect negligible differences between atomic relaxation of the magnetic adatoms on Nb(110) depending on the GGA or LDA functional.
We included a comment on this in Section 2, and extended the end of Section 1 in the revised version of the manuscript.
Reviewer 2:
- The number of layers of Nb seems to be rather small (at least in the VASP calculation) and it is probably not converged. The authors should explain why they used such numbers and if the results are properly converged.
Authors' reply:
Since the purpose of using VASP was to determine the geometrical relaxation of the magnetic adatoms on the Nb(110) substrate, the precise number of the Nb atomic layers in the slabs is not that critical. This is evidenced by our comparison with Ref. [8] concerning Fe adatom on Nb(110), which is provided in Section 2.1, where we find very similar values for the atomic relaxations: 175 (our) and 170 pm (Ref. [8]) (here the definition of the relaxation might also differ), and for the spin moments of Fe: 2.14 (our) and 2.2 Bohr magneton (Ref. [8]). We used 4 atomic layers ~0.7 nm thick Nb slab, and Ref. [8] used ~2.5 nm thick Nb slab (but the size of the surface cell is not reported there). The Mn/Nb(110) relaxation was already published in Ref. [7]. Furthermore, note that in the KKR calculations (from which all the physical results are derived) a semi-infinite Nb substrate is considered (Section 2.2: "The Nb(110) surface has been modeled as an interface region between semi-infinite bulk Nb and vacuum..."), so the validity of the results from this aspect cannot be questioned.
Reviewer 2:
- The argument that the superconducting state might not affect the magnetic patterns is not convincing and it does not imply that the same results would be valid on a superconducting surface, where the electronic structure and interaction between the surface and atoms might be different. Can the authors elaborate on that?
Authors' reply:
Indeed, in the Introduction of our manuscript we make the assumption, that the superconductivity does not affect the magnetic pattern. While we think that the energy scale argument we mentioned to support this assumption is qualitatively correct, a more concrete example is provided in Fig. 3(b) of our manuscript. This figure shows that for both spin channels the main changes in the electronic structure between the FM- and AFM-coupled Mn u-1NN dimer happen well below the Fermi level and the band energy difference as a function of the upper integration limit, see Eq. (9), appears quite stable above approximately EF-0.3 eV. This implies that a superconducting gap of about 1-2 meV at EF shouldn’t affect the band energy difference between the FM and AFM configurations. From the good agreement of the data in Table 2 and Table 3, it follows that the above statement applies to the magnetic coupling for this dimer, too. Since the integrand of the integral on the rhs of Eq. (9) contains (e-EF), the changes close to the Fermi level are suppressed, therefore, we expect that the above reasoning is generally valid for the magnetic couplings.
We also emphasize that in Ref. [7] the magnetic structure of Mn dimers deposited on Nb(110) as reported in the present manuscript was used to calculate the Yu-Shiba-Rusinov states in the superconducting phase and remarkable agreement with the experimental observations was found. Though quite indirectly, this also supports our assumption in question.
Reviewer 2:
- Although in general the physical explanations are correct, there is in some cases a lack of physical insight and the authors simply describe what they find (e.g. values of the anisotropy tensor in page 10). The authors should delve more into the physics in general.
Authors' reply:
Despite a huge amount of experimental and theoretical attempts, the precise microscopic origin of the magnetic interactions of relativistic origin, such as the magnetic anisotropy and the DMI, in magnetic nanostructures remained a mostly unexplored scientific task. There are general arguments, like the strength of the SOC or the spin-polarizability of the substrate, which support using heavy transition metals to enhance these interactions, but a very detailed materials specific explanation how these interactions are being formed is possible only via ab initio calculations. This was the main goal of our presented work. At best, we attempted to give a physical insight to explain the characteristics of the isotropic magnetic couplings in atomic dimers and in selected atomic chains in terms of their electron orbital decomposition, and related them to the orbital-decomposed LDOS and to the band-energy differences between the FM and AFM configurations.
We also showed in our paper that the anisotropy tensor can also result in the formation of non-collinear magnetic states.
These two aspects have never been reported in the literature before, due to our knowledge, thus we hope that our study sheds some new light on the physics behind the magnetic couplings and the forming of non-collinear magnetic states in these systems.
We emphasized the new physical insights in the Abstract provided by our work.
Reviewer 3 Report
The paper "Electronic and Magnetic Properties of Building Blocks of Mn and Fe Atomic Chains on Nb(110)" reports interesting results about spin order and the properties of magnetic atoms on surfaces. The manuscript is well written and presented in a logical manner. The authors seem to ignore, though, several of the advances made on molecular metallic chains (e.g. single chain magnets, Gd-radical chains, etc...) where chiral order has been , for example, demonstrated. I would suggest expanding the scope of the paper in the intro and conclusions.
Author Response
Reviewer 3:
The paper "Electronic and Magnetic Properties of Building Blocks of Mn and Fe Atomic Chains on Nb(110)" reports interesting results about spin order and the properties of magnetic atoms on surfaces. The manuscript is well written and presented in a logical manner. The authors seem to ignore, though, several of the advances made on molecular metallic chains (e.g. single chain magnets, Gd-radical chains, etc...) where chiral order has been, for example, demonstrated. I would suggest expanding the scope of the paper in the intro and conclusions.
Authors' reply:
Following this suggestion for the extension of the scope and widening the context of our research, we extended the discussion in the Introduction and in the Summary to acknowledge the important field of the mentioned magnetic molecular chains including examples from the literature.